# Comparison between Gradual Reduced Nicotine Content and Usual Nicotine Content Groups on Subjective Cigarette Ratings in a Randomized Double-Blind Trial

**DOI:** 10.3390/ijerph17197047

**Published:** 2020-09-26

**Authors:** Wenxue Lin, Nicolle M. Krebs, Junjia Zhu, Jonathan Foulds, Kimberly Horn, Joshua E. Muscat

**Affiliations:** 1Department of Public Health Sciences, Pennsylvania State University College of Medicine, Hershey, PA 17033, USA; nkrebs@pennstatehealth.psu.edu (N.M.K.); jzhu2@phs.psu.edu (J.Z.); jfoulds@pennstatehealth.psu.edu (J.F.); jmuscat@pennstatehealth.psu.edu (J.E.M.); 2Department of Population Health Sciences, Virginia Tech Carilion Research Institute, Roanoke, VA 24016, USA; kahorn1@vt.edu

**Keywords:** reduced nicotine content, usual nicotine content, modified cigarette evaluation questionnaire, cigarette-liking scale

## Abstract

In 2018, the United States Food and Drug Administration (FDA) issued an advanced notice of proposed rulemaking to reduce nicotine in tobacco products to produce a minimally addictive or nonaddictive effect, but there was a research gap in the subjective responses of reduced-nicotine-content cigarettes. We compared the responses of the modified cigarette evaluation questionnaire (mCEQ) and cigarette-liking scale (CLS) between the gradually reduced nicotine content (RNC) group and the usual nicotine content (UNC) group. Linear mixed-effects models for repeated measures were used to analyze and compare the change over time for the mCEQ and CLS across the two treatment groups (RNC and UNC). We found that the change over time for the mCEQ and CLS was significant between the RNC and the UNC treatment groups at the beginning of visit 6 with 1.4 mg nicotine/cigarette. At visits 8 and 9, the RNC group reported significantly lower satisfaction scores compared to UNC. Subscale analysis showed that smoking satisfaction decreased in RNC while other measures, such as cigarette enjoyment, did not change. Understanding the impact of nicotine reduction on cigarette subjective responses through evaluation and liking scales would provide valuable information to the FDA on nicotine reduction policies for cigarettes.

## 1. Introduction

Cigarette smoking in the United States (U.S.) has decreased from 42.4% in 1964 to 13.7% in 2018; nonetheless, cigarette smoking remains the primary preventable cause of death in the U.S., being responsible for more than 480,000 annual deaths [1,2,3]. Although smoking-related diseases are caused by chemicals in tobacco or created during the combustion process, nicotine is responsible for the addiction [3]. Prior research and literature demonstrated that a low-nicotine product standard for cigarettes could positively affect public health, specifically, implementation of a nicotine reduction policy could prevent 16 million people from smoking by 2060 and prevent approximately 8.5 million tobacco-related deaths in the United States by 2100 [4]. The U.S. Food and Drug Administration (FDA) issued an Advanced Notice of Proposed Rulemaking (ANPRM) in 2018 to reduce nicotine in tobacco products to make them minimally addictive or nonaddictive [5]. Clinical studies have indicated the benefit of very low nicotine content (VLNC) in the reduction of nicotine exposure levels, biomarkers of smoke toxicant exposure, lower dependence, fewer cigarettes smoked per day, and increased quit attempts [6,7,8,9,10,11,12].

One consideration is the impact of the method of nicotine reduction (gradual vs. immediate) in cigarettes [13]. Although an immediate transition to VLNC cigarettes in a recent study by Hatsukami et al. resulted in better health outcomes [8], it led to a higher rate of participant attrition and lower compliance compared to gradual reduction [8]. The method of nicotine reduction is an important factor. Immediate reduction may offer less harmful exposure overall, but gradual reduction may garner more support and compliance from smokers.

Subjective measures are important tools for measuring the acceptability and sensory effects of low nicotine products. Acceptability may modify the behavioral responses to reduced nicotine products used in trials, such as participant compliance and dropout. This could foreshadow the population-level response to a reduced nicotine policy. A secondary analysis in the Hatuskami et al. trial found that the immediate nicotine reduction group scored significantly lower than the gradual reduction group on multiple subscales of the modified cigarette evaluation questionnaire (mCEQ), which measures both positive and negative reinforcements of smoking [13]. Other studies have examined cigarette characteristics (such as taste and draw) by measuring the subjective effects of smoking, recognizing that nicotine alone is not the only contributor to the smoking experience [14].

As a national nicotine reduction policy has yet to be administered, it is still important and necessary to explore the gradual reduction method. Few studies compared the multiple factors of the smoking experience between the gradually reduced nicotine content (RNC) cigarettes and usual nicotine content (UNC) cigarettes in a randomized controlled trial. We aimed to study and compare the change over time using the mCEQ [15] and the cigarette-liking scale (CLS) [14] across two treatment groups (RNC and UNC). This study included a study population of low socioeconomic status (SES) smokers. Unique subpopulations, such as low SES smokers, have higher levels of nicotine dependence and increased smoking rates, which may cause different experiences with reduced nicotine cigarettes, such as higher aversions and lower acceptability scores.

## 2. Materials and Methods

A two-site, two-arm, double-blind, parallel-group, randomized clinical trial was conducted at the Penn State College of Medicine (Hershey, PA, USA) and George Washington University (Washington, DC, USA) between 2015 and 2018. The study consisted of 4 phases (baseline 1, baseline 2, randomized, and treatment choice) with 11 clinical visits at the study centers over 33 weeks [16]. Baseline 1 (visit 1) included smoking their usual brand cigarettes for 1 week. Baseline 2 (visits 2 and 3) included 2 weeks of smoking UNC study cigarettes (nicotine content approximately 11.6 mg/cigarette). In the 18-week randomized phase, participants were randomized to the control arm in which they continued on the UNC study cigarettes, or the intervention arm to receive progressively RNC study cigarettes [16]. Participants in the control group received SPECTRUM cigarettes with 11.6 mg nicotine/cigarette for the entire 18-week period. Participants in the gradual group received SPECTRUM cigarettes with nicotine contents of 7.4 (visit 4), 3.3 (visit 5), 1.4 (visit 6), 0.7 (visit 7), and 0.2 (visits 8 and 9) mg nicotine/cigarette [16]. The randomized phase was followed by a 12-week treatment choice phase (visits 10 and 11). The study flow diagram is shown in Appendix A. The detailed study protocol, methods, randomization procedures, and primary outcomes of the trial were published previously [16]. During the study visits, multiple measures were assessed, such as exhaled carbon monoxide and the Fagerström test for cigarette dependence (FTCD) [17]. In this study, participants completed the mCEQ and CLS questionnaires from visits 3 to 9. Detailed information about the mCEQ and the scoring of subscales are available [15]. The CLS includes questions from a previously published scale [14,18]. Demographic and smoking characteristics of randomized participants are included in Appendix A. All subjects provided written inform consent and the study was approved by the George Washington University (IRB #011507) and Penn State College of Medicine (STUDY #00000660) Institutional Review Boards. The study is registered at clinicaltrials.gov (NCT01928719).

### Statistical Methods

Linear mixed-effects models for repeated measures were used to analyze and compare the change over time for the mCEQ and CLS across two treatment groups (RNC and UNC). The first-order autoregressive (AR(1)) structure was assumed for repeated measures in all models. The linear mixed models were used to evaluate the effects of discrete time (visits 4–9), groups (RNC vs. UNC group), and the time-by-group interaction, while adjusting for the baseline (visit 3) measure of the outcome. Known confounders (e.g., sex, site, and menthol flavor) were consistently included in the adjusted models, and other covariates were included since they may be possible confounders (age group, education, cigarette brand, race, FTCD score, cotinine, and employment status) [13]. Estimated least-squares mean differences and corresponding 95% confidence intervals (CIs) were reported from visits 4 to 9. Study data were collected, managed, and downloaded from the Research Electronic Data Capture (REDCap) hosted at the Penn State College of Medicine [19]. A total of 245 participants completed the study: 122 in the reduced-nicotine level group and 123 in the usual-nicotine level group. Analyses were conducted by the Penn State TCORS Biostatistics and Database Management Core using statistical software SAS Version 9.4 (SAS Institute, Cary, NC, USA). All tests were two-sided at the 0.05 significance level.

## 3. Results

### 3.1. Modified Cigarette Evaluation Questionnaire (mCEQ)

Figure 1 shows the comparison of mean scores between RNC and UNC on each of the five mCEQ subscales from visits 4 to 9. Table 1 lists the mCEQ subscales mean differences between the RNC and UNC groups with *p*-values. For the satisfaction subscale, the rating score for RNC was higher than UNC at visits 4 and 5 (RNC nicotine content: 7.4 and 3.3 mg/cigarette, respectively), but starting from visit 6 (nicotine content: 1.4 mg/cigarette), RNC showed a significantly lower rating score in satisfaction compared to UNC (Table 1). For the psychological reward subscale, the rating score showed no significant difference between RNC and UNC until visit 9 (*p* = 0.0192, Table 1). For the enjoyment of respiratory tract sensations, a significant difference between RNC and UNC was present at visit 7 (*p* = 0.0486, Table 1), while the following comparison indicated no difference at visits 8 and 9. For craving relief, significant differences were detected at visits 6 and 8 (*p* = 0.0072, and 0.0388, respectively, Table 1). As for aversion, no significant differences were observed between the two treatment groups from visits 4 to 9 (*p* > 0.05). The mCEQ subscales mean differences between the RNC and UNC groups with 95% CIs are included in Appendix A.

### 3.2. Cigarette-Liking Scale

Figure 2 depicts the comparison between RNC and UNC on each of the 10 items in the CLS from visits 4 to 9. Table 2 lists the CLS mean differences between the RNC and the UNC groups with *p*-values. For CLS 1 (strength), comparison results indicated significant rating score differences between RNC and UNC from visits 4 to 9 (Table 2). The mean rating score for RNC decreased from visits 4 to 9, with the widest gap between the two treatment groups from visits 6 to 9 (Figure 2, CLS item 1). For CLS 5 (taste), the RNC group showed a significantly lower rating than the UNC group for all of the visits except visit 4. The mean difference between the two groups was the highest at visit 9 (Figure 2). For CLS 6 (satisfaction), significant differences were detected at visits 5, 6, 8, and 9. As for CLS 8 (likelihood of buying cigarettes) and CLS 9 (nicotine from cigarettes), the RNC group rated significantly lower from visits 6 to 9 compared to the UNC group (Table 2, Figure 2). For CLS 4 (harshness), CLS 7 (tobacco vs. just air), and CLS 10 (satisfying hit), three out of six visits showed significantly lower scores among the RNC group compared to the UNC group. For both CLS 7 and CLS 10, the difference between the two groups became significant at visits 6, 8, and 9, whereas CLS 4 became significant at visits 5, 8, and 9. For CLS 2 (heat), none of the visits showed significantly different scores between the two groups except visit 5. Lastly, for CLS 3 (hard to draw), the RNC and the UNC groups indicated no difference at any visit. The CLS mean differences between the RNC and UNC groups with 95% CIs are included in Appendix A.

## 4. Discussion

Our mCEQ results indicated that gradual reduction in nicotine impacts the positive reinforcements of smoking (i.e., smoking satisfaction) versus negative reinforcements (i.e., aversion). These findings are consistent with prior research conducted by Smith et al., where the satisfaction and psychological reward presented significant differences between gradual reduction and control groups and no differences were observed in aversion subscales [13]. They found a significant difference between the gradual reduction group and control group in both the enjoyment of respiratory tract sensations and in craving relief [13], whereas our study showed no difference for the RNC and UNC groups in rating these two mCEQ subscales at visit 9. These results indicated that smokers still enjoy some of the sensory aspects of smoking, despite lowered nicotine content.

Prior laboratory-based research suggested that it is relatively hard for smokers to discriminate VLNC cigarettes from other low nicotine cigarettes, but more often can discriminate between VLNC and UNC cigarettes [20]. In our study, based on CLS 1 (strength of the cigarette) and CLS 9 (how much nicotine thought to be in these cigarettes compared to usual cigarettes), smokers were able to distinguish the lowered nicotine doses in a dose–response fashion. Other cigarette characteristic ratings, such as taste, was consistently rated lower among the RNC group. The lower rating of taste could contribute to a lower abuse liability of RNC cigarettes. The RNC cigarettes were consistently rated lower on CLS 6, 8, and 9 given four out of six visits demonstrated significant results. The maximum permissible limit for nicotine content under a new nicotine standard will be at, or near, minimal or non-addicting levels.

Overall, visit 6 was usually the starting point of significant differences for some mCEQ and CLS measures between the two treatments. The nicotine content for this visit was 1.4 mg nicotine per cigarette. Visit 6 or 1.4 mg nicotine per cigarette may play a vital role in participants’ ratings on the liking scale. In light of current research, VLNC is usually defined as less than 0.6 mg nicotine per cigarette. The findings from our study revealed that more consistent significant changes in perception of key cigarette effects (e.g., satisfaction, craving reduction and likelihood of buying) are evident at cigarette nicotine content of 1.4 mg and below.

This study has several limitations. First, we used SPECTRUM research cigarettes (22nd Century Group, Inc., Williamsville, NY, USA), obtained from the National Institute of Drug Abuse’s Drug Supply Program. Commercially-made RNC and VLNC cigarettes may be different from our research cigarettes including design and additives [13]. In other words, the generalizability of subjective responses from VLNC research cigarettes and commercial cigarettes may be reduced [13]. Second, our study was a double-blind, randomized clinical trial, which cannot reflect the real-world situation since smokers will be aware of the reduced nicotine content from the cigarette package if the FDA decides to implement a reduced nicotine policy [21]. Additionally, our study was conducted in a pre-policy environment where participant’s usual brand cigarettes were still available, providing the opportunity for non-compliance with only smoking the research cigarettes.

## 5. Conclusions

Findings from our study revealed that consistent significant differences between the RNC and the UNC treatment groups started to occur at visit 6 with 1.4 mg nicotine/cigarette. The most consistent differences occurred at visits 8 and 9 when the RNC group was smoking VLNC cigarettes (0.2 mg nicotine/cigarette). The study indicates that even with reduced nicotine, cigarettes still provide some pleasurable and reinforcing effects. Subjects using RNC stated they were less likely to purchase RNC cigarettes, which would facilitate the goal of a nicotine reduction policy, but also points to the need to reduce access to black market high nicotine cigarettes. Understanding the impact of nicotine reduction on subjective effects through cigarette evaluation and liking scales would provide valuable information and possible direction to the FDA on implementing a nicotine reduction policy for cigarettes.

## Figures and Tables

**Figure 1 ijerph-17-07047-f001:**
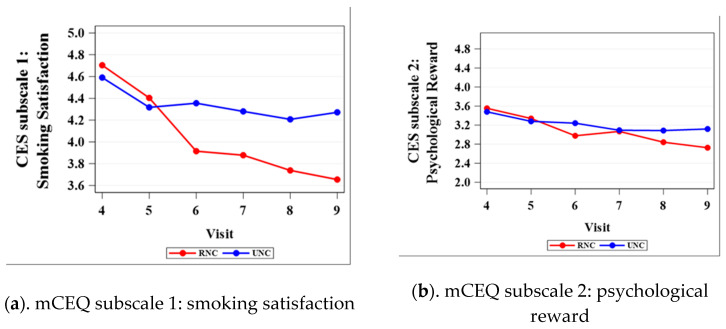
Mean scores of RNC (reduced nicotine content) and UNC (usual nicotine content) on each of the five mCEQ (modified cigarette evaluation questionnaire) subscales from visits 4 to 9.

**Figure 2 ijerph-17-07047-f002:**
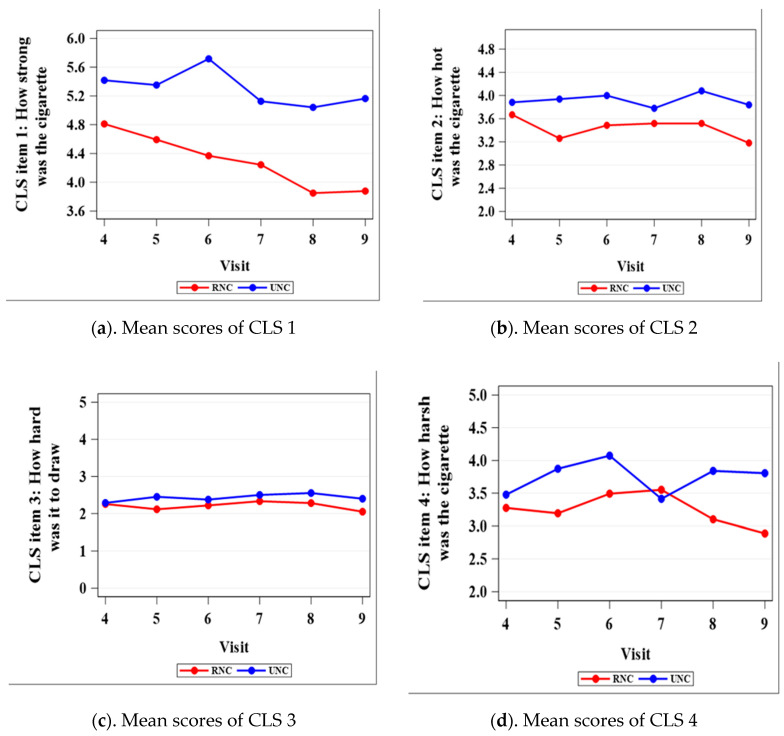
Mean scores of RNC and UNC on each of the 11 CLS (cigarette-liking scale) items from visits 4 to 9.

**Table 1 ijerph-17-07047-t001:** Results of analyses comparing mCEQ subscales in RNC vs. UNC groups.

No. of Visit	Visit 4	Visit 5	Visit 6	Visit 7	Visit 8	Visit 9
Nicotine Content	7.4 mg	3.3 mg	1.4 mg	0.7 mg	0.2 mg	0.2 mg
	*p*-Value	*p*-Value	*p*-Value	*p*-Value	*p*-Value	*p*-Value
Satisfaction ^1^	0.5214	0.6304	**0.0172**	**0.0356**	**0.0181**	**0.0026**
Psychological reward ^2^	0.6113	0.6874	0.0809	0.8768	0.1326	**0.0192**
Aversion ^3^	0.7556	0.7991	0.8210	0.3690	0.6835	0.9083
Enjoyment of Respiratory Tract Sensations ^4^	0.7142	0.4995	0.3161	**0.0486**	0.8528	0.5261
Craving reduction ^5^	0.4241	0.5367	**0.0072**	0.4021	**0.0388**	0.5260

Note: Bolded *p*-values indicate significance. Linear mixed-effects models adjusted for baseline (visit 3) measure of the outcome, flavor, site, age group, education group, brand, sex, race, FTCD score, cotinine, and employment status. 1: mCEQ subscale 1; 2: mCEQ subscale 2; 3: mCEQ subscale 3; 4: mCEQ subscale 4; 5: mCEQ subscale 5. Unit for Nicotine Content: mg/cigarette. mCEQ: modified cigarette evaluation questionnaire; RNC: reduced nicotine content; UNC: usual nicotine content.

**Table 2 ijerph-17-07047-t002:** Results of analyses comparing CLS in RNC vs. UNC groups.

	Visit 4	Visit 5	Visit 6	Visit 7	Visit 8	Visit 9
	*p*-Value	*p*-Value	*p*-Value	*p*-Value	*p*-Value	*p*-Value
CLS 1	**0.0386**	**0.0114**	**<0.0001**	**0.0055**	**0.0003**	**0.0002**
CLS 2	0.4658	**0.0235**	0.0935	0.4080	0.0863	0.0516
CLS 3	0.9012	0.1868	0.5481	0.5340	0.3355	0.2267
CLS 4	0.5187	**0.0343**	0.0777	0.6848	**0.0371**	**0.0119**
CLS 5	0.0593	**0.0125**	**<0.0001**	**0.0005**	**0.0018**	**<0.0001**
CLS 6	0.4326	**0.0290**	**<0.0001**	0.1938	**0.0135**	**<0.0001**
CLS 7	0.6193	0.4824	**0.0042**	0.1763	**0.0010**	**0.0069**
CLS 8	0.5096	0.1962	**0.0082**	**0.0054**	**0.0016**	**0.0002**
CLS 9	0.1305	0.0545	**<0.0001**	**0.0080**	**<0.0001**	**<0.0001**
CLS 10	0.8300	0.3574	**0.0130**	0.0621	**0.0033**	**0.0054**

Note: Bolded p values indicate significance. Linear mixed-effects models adjusted for baseline (visit 3) measure of the outcome, flavor, site, age group, education group, brand, sex, race, FTCD score, cotinine, and employment status. CLS items: CLS item 1: How strong was the cigarette? (1 = not at all, 10 = extremely); CLS item 2: How hot was the cigarette? (1 = not at all, 10 = extremely); CLS item 3: How hard was it to draw? (1 = not at all, 10 = extremely); CLS item 4: How harsh was the cigarette? (1 = not at all,10 = extremely); CLS item 5: How much taste did you get from the cigarette? (1 = not at all, 10 = extremely); CLS item 6: How satisfying was the cigarette? (1 = not at all, 10 = extremely); CLS item 7: How much tobacco vs. ‘just air’ did you get from the cigarette? (1 = just air, 10 = just tobacco); CLS item 8: What is the likelihood that you would buy cigarettes like these? (1 = not at all, 10 = extremely); CLS item 9: How much nicotine do you think these cigarettes gave you compared to your usual cigarettes? (1 = much less, 5 = much more); CLS item 10: How satisfying was the hit these cigarettes gave you compared to your usual cigarettes? (1 = much less, 5 = much more).

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
