# Peer review of "Comparison between Gradual Reduced Nicotine Content and Usual Nicotine Content Groups on Subjective Cigarette Ratings in a Randomized Double-Blind Trial"

_ijerph, 2020, doi:10.3390/ijerph17197047_

Round 1
Reviewer 1 Report
Dear Authors
1. It was with great pleasure that I read your article.
This paper is very useful in pointing out key differences between, Gradual Reduced Nicotine Content (RNC) group and the Usual Nicotine Content (UNC) group on Subjective Cigarette Ratings in a Randomized Double-blind Trial. The article with a very simple method proved the rightness of the goals, and at the same time indirectly affects the health of smokers and their complications
The presented data may be crucial for a full understanding of nicotine reduction behaviors, especially among smokers who are addicted to nicotine.
2. I would have expected better charts with standard deviations
3. Lack of informed consent to participate in the study, the more so as patients received lower doses of nicotine and this could have an adverse effect on the patient
4. Lack of information and consent from the bioethics committee
5. The Y line of the charts - poorly legible (larger type indicated)
6. Graphs should contain standard deviations
Best regards
Reviewer 2 Report
Lin et al. Comparison between gradual reduced nicotine……….
This paper addresses an issue of actuality since FDA has shown interest in the concept of severely limiting the nicotine content in cigarettes. The study looks as the effects of smoking reduce nicotine content cigarettes to cigarettes with usual nicotine content and implements the reduced nicotine cigarettes gradually.
The design is straightforward and the way the authors report the results is short and consistent.
My comments for improving the paper are the following.
- The results from the CEQ (one of the dependent variables) are nicely and clearly shown in fig 1. In Table 1 are the magnitude of the differences reported digitally. There is a lot of overlapping and volume consumption with both Fig 1 and Table 1.
The same reasoning goes for CLS and Fig. and Table 2.
- In Materials and Methods it says that the study took 33 weeks (line74). When I add up the 2 weeks on smoking usual nicotine content and the 18 weeks on randomised treatment it adds up to only 20 weeks. Please clarify! Also number of visits is 11 but in the figures there is only 9 visits. What happened during visit 10 and 11? Clarify!
- The Fagerstrom Test for Nicotine Dependence (FTND) has been renamed to the Fagerstrom Test for Cigarette Dependence (FTCD).
- Line 34 “Smoking related diseases are caused by chemical in tobacco….”. Should it be chemicals?
- Line 38, should “death” be deaths?
- In the abstract CES (cigarette evaluation scale) is used but throughout the rest of the paper CEQ (cigarette evaluation questionnaire) is used. Be consistent!
Round 2
Reviewer 2 Report
The authors have adequately dealt with my comments